# Influence of the Growth Ambience on the Localized Phase Separation and Electrical Conductivity in SrRuO₃ Oxide Films

**Hsin-Ming Cheng**

Organic Electronics Research Center and Department of Electronic Engineering, Ming Chi University of Technology, New Taipei City 24301, Taiwan; SMCheng@mail.mcut.edu.tw

**Abstract:** Perovskite SrRuO$_3$ (SRO) epitaxial thin films grown on SrTiO$_3$ (STO) (001) have been synthesized using pulsed laser deposition (PLD) under a series of oxygen pressures. High quality and conductive SRO thin films on STO have been achieved at $10^{-1}$ Torr oxygen pressure with the epitaxial relation of (110)<001>$_{SrRuO_3}$//(001)<010>$_{SrTiO_3}$. The lattice parameters of the thin films exhibit huge expansion by reducing the ambience (~$10^{-7}$ Torr) during deposition, and the resistance increases by about two orders higher as compared with the low oxide pressure ones. The rise of resistivity can be ascribed to not only the deficiency of Ru elements but also the phase transformation inside SRO thin films. The correlation of growth ambience on the structural transition and corresponding resistivity of epitaxial oxide thin films have been explicitly investigated.

**Keywords:** pulsed laser deposition; functional oxide; phase transformation; electrical conductivity

## 1. Introduction

Complex ruthenium oxide materials are fascinating because they possess a range of physical properties including being superconductive [1,2] and ferromagnetic [3–11], and they also have metallic properties [12–15]. Among the ruthenium oxide family, metallic oxide SrRuO$_3$ (SRO) has attracted much attention because of its functionality as a conducting electrode in the integration of perovskite oxide devices such as magnetic tunnel junctions [16,17]. SRO, a 4$d$ transition metal oxide with a Curie temperature of 161 K, is a Pbnm orthorhombic structure with lattice parameters $a$ = 5.57 Å, $b$ = 5.53 Å, and $c$ = 7.86 Å [18]. Due to a small distortion with respect to the ideal cubic structure, SRO is also regarded as pseudo-cubic with a parameter of 3.93 Å [19,20]. In addition, the conductivity of SRO comes from a π* narrow-type conduction band, which originates from a strong hybridization of oxygen 2$p$-derived states with ruthenium $d$ states [21,22]. The resistivity of SRO thin film is about 200 μΩ-cm at room temperature (RT) [15]. Structural compatibility with perovskite oxides and metallic conductivity make SRO widely used for electrodes in oxide-based devices.

As an itinerant ferromagnetic oxide, in the past twenty-five years, SRO thin films have been proposed that can be successfully deposited on vicinal substrates such as SrTiO$_3$ (STO) through heteroepitaxial step-flow growth [23–26]. This model has offered a rational explanation for the morphological phase diagram, step bunching, and island formation of SRO surfaces. However, in order to obtain high-quality functional oxide films, the control of the atmosphere, especially the oxygen ambience, plays an essential role in the aspects of defects [6,27], stoichiometry [14], oxidations [28], and reductions [28]. Furthermore, oxygen partial pressure dramatically causes phase transformations as well [7,11–13].

For SRO functional oxide materials, the ambience effects on their surface stability have been studied by post-annealing at temperatures higher than 600 °C under different ambiences [28,29].

Lee et al. reported that SRO will decompose below an oxygen pressure of $10^{-4}$ Torr at 720 °C [29]. Shin et al. further listed the decomposition reactions of SRO at a range of oxygen pressures and temperatures based on thermodynamic theory [28]. Although these reactions are complicated due to the fact that nonstoichiometric SrO will not coexist in equilibrium with SRO, the report also pointed out that ruthenium seems to lose more than strontium at high vacuum ($10^{-7}$ Torr) and high temperature (700 °C). Siemons et al. reported that either too low or too high oxygen activity could lead to the deficiency of Ru in SRO thin films, which increases the resistivity [14]. On the other hand, Lee et al. recently reported a close correlation between the phase transitions and oxygen evolution reaction of SRO epitaxial thin films by systematically introducing Ru–O vacancies [30]. However, even though previous reports have revealed the resistivity of SRO thin films as entire thin film, namely the macro-scale measurement, there are few investigations that mention the microstructural and local electrical properties of SRO with various growth ambience. As a result, to understand the complexity between the process and the conductivity that can be used for practical application in the future, the subject related to the growth ambience of SRO thin films needs to be diligently discussed. In this study, we control the oxygen pressure from $10^{-1}$ to $10^{-7}$ Torr in order to systematically study the influence of growth ambience on the structure and conductivity of SRO thin films. The crystallization and the corresponding conductivity of SRO thin films vary significantly as a function of the ambient environments. Furthermore, the variations of crystallization from the deficiency of Ru elements, surface morphology, degree of stoichiometry, and the microstructure caused by phase transformation in SRO thin films are also investigated.

## 2. Experiment

Epitaxial thin films of SRO were deposited on SrTiO$_3$ (001) substrate using pulsed laser deposition (PLD, DCA PLD500, Turku, Finland) with a target to substrate distance of ~55 mm using a KrF excimer laser ($\lambda$ = 248 nm, pulse duration 4 ns). The samples were grown in a vacuum chamber with a background pressure of $10^{-8}$ Torr at 700 °C using pulsed energy of 90 mJ with a repetition rate of 8 Hz. The fluence was refined with an attenuator to get 6 J/cm$^2$ on the target, with a laser spot size of 1.5 mm$^2$. The thickness of the SRO thin films was controlled to around 300 nm. After growth, the samples were placed in the original position and remained there to cool naturally. The ambient oxygen during growth was varied from $10^{-1}$ to $10^{-7}$ Torr, while other conditions were kept the same in order to confirm how oxygen pressure affected the structural and electrical properties. The crystal structure of the grown films was identified using a high-resolution four-circle X-ray diffractometer (Bede D1, HRXRD, Bede, Durham, UK) with Cu-K$\alpha_1$ radiation. The surface morphology of the samples was characterized by field emission scanning electron microscopy (FESEM, JEOL-6500, Tokyo, Japan), and their ingredients were analyzed using an energy dispersive spectrometer (EDS) operated at 5 kV. Cross sectional TEM specimens were prepared by focused ion beam (FIB) and characterized by field emission transmission electron microscope (FETEM, JEOL JEM-2100F) operated at 200 kV. The electrical resistivity at RT was measured using a standard four-probe technique. Atomic force microscopy (AFM, Bruker Innova, SPM, Billerica, MA, USA), working in contact mode, was used to characterize the surfaces of the films. Local electrical conductivity was measured using high-resolution conductive AFM (C-AFM) by applying a bias voltage of 0.2 V between the C-AFM tip and the surrounding silver colloid.

## 3. Results and Discussion

X-ray reciprocal space mapping (RSM) along the $\theta$–$2\theta$ scan and the rocking curve $\theta$ around STO (002) and SRO (220) reflections were used to analyze the structure of SRO on STO (001). Figure 1 shows the RSM results for SRO thin films grown under an oxygen pressure of $10^{-1}$ Torr (Sample A) and $10^{-6}$ Torr (Sample C), respectively. Only SRO (220) and STO (002) reflections are observed, revealing that the SRO (110) thin films are parallel to the STO (001). These results are consistent with previous reports. As can be seen in Figure 1, Sample A is different from Sample C in both the peak position and the full width at half maximum (FWHM) of the $\theta$-rocking curve along the surface normal direction. The peak

position and FWHM of Sample A are 46.169° and 0.074°, respectively. These results indicate that SRO thin films have a $d_{110}$-spacing of 3.928 Å, for which the value is very close to its bulk value (3.925 Å). The sharp rocking curve also reveals an excellent crystallization of the SRO thin films.

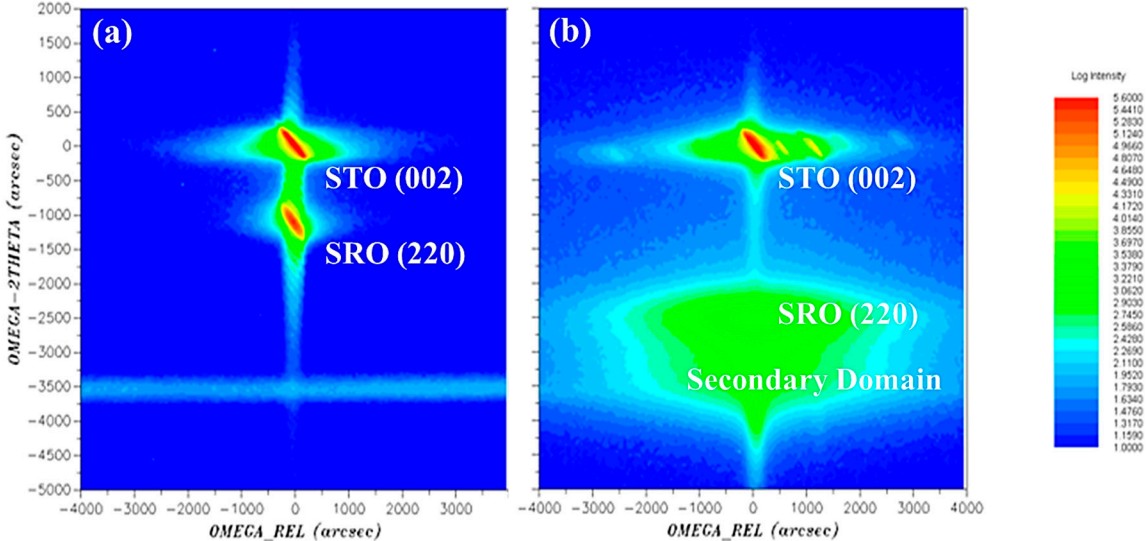

**Figure 1.** RSM around the normal surface of (002) reflection for SrTiO$_3$ (STO) and (220) reflection for SrRuO$_3$ (SRO) thin films of (**a**) Sample A and (**b**) Sample C, grown at $10^{-1}$ and $10^{-6}$ Torr, respectively. (The unit of arcsec is equal to 1 over 3600 degree).

Figure 2 illustrates the ϕ-scans across the off-normal SRO (112) and STO (011) reflections of Sample A. It reveals a sharp width of 0.0471° (0.0135° for STO substrate) with a near four-fold symmetry for SRO (112). It indicates a very high quality of SRO thin film by using the high oxygen pressure of $10^{-1}$ Torr. Furthermore, the results point out that the in-plane epitaxial relationship follows $\{001\}_{\mathrm{SrRuO_3}}\|\{010\}_{\mathrm{SrTiO_3}}$, which is consistent with the work of Kan et al. [14].

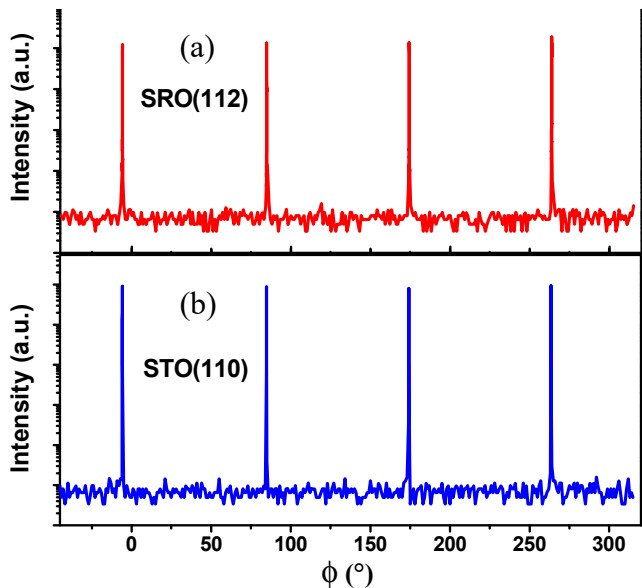

**Figure 2.** The ϕ-scans across the off-normal SrRuO$_3$ reflection (**a**) and SrTiO$_3$ reflection (**b**) of Sample A.

On the contrary, X-ray diffraction also shows that the lattice parameter and FWHM expand a lot in Sample **C**, which was grown under a lower oxygen pressure. The calculated lattice parameter and

rocking curve of Sample **C** are 3.959 Å and 0.2725°, respectively. Consequently, the results indicate that the crystal structure of SRO thin films is very sensitive to oxygen pressure during growth. In this work, it was found that a high oxygen pressure of about 100 mTorr facilitated the growth of high crystal-quality SRO thin films. At the lower side of Figure 1b, a small elliptical region is found in Sample C, which indicates that SRO is composed of two domains with different lattice constants (3.959 and 3.987 Å). Sample D also shows the multi-domain phenomenon, while the lattice constants of sample D are larger than those of Sample C, as shown in Figure S1. These secondary domains might be formed due to the instability of SRO at high temperature and simultaneously high vacuum, and they would be almost undetectable while the oxygen pressure is above $10^{-4}$ Torr.

The detailed variations of lattice constants, strains, and sheet resistances under different growth oxygen pressures are listed in Table 1. The out-of-plane lattice constant expands as the oxygen pressure decreases, and the secondary domains with larger lattice parameters are clearly observed bellow a growth ambience of $10^{-5}$ Torr. Furthermore, the electrical property changes with the growth ambience. From the results of the four-probe method, the resistivity of Sample B is slightly high but still of the same order as Sample A. However, the resistivity of Samples C and D substantially increase with decreasing oxygen pressure during growth. Sample D, grown without the addition of oxygen, reveals a resistivity about two orders higher than that of Sample A. Therefore, structure and electricity seem to vary with growth ambience in two stages. The lattice constant of SRO thin films increases, but the conductivity decreases gradually while controlling the oxygen pressure from $10^{-1}$ to $10^{-4}$ Torr. Several secondary domains are formed, and the conductivity of SRO thin films drops while the oxygen pressure is below $10^{-4}$ Torr. Therefore, there should be a mechanism responsible for the different variations in the two stages. The mechanism of differences in the decrease rate is discussed in detail below.

**Table 1.** The lattice parameters, strains, and resistivity of SRO thin films fabricated under different oxygen pressures.

| Sample | A | B | C | D |
|---|---|---|---|---|
| Oxygen pressure (Torr) | $10^{-1}$ | $10^{-4}$ | $10^{-6}$ | $10^{-7}$ |
| *d*-spacing of (110) (Å) | 3.928 | 3.947 | 3.959 | 3.963 |
| Secondary domain *d* (Å) | – | – | 3.987 | 4.076 |
| FWHM (Degree) | 0.0741 | 0.2725 | 0.3190 | 0.4045 |
| Resistivity (μΩ-cm) | 224 | 600 | 4380 | 61,440 |

*3.1. SEM*

The morphologies of SRO thin films were measured by SEM, as shown in Figure 3. As can be seen, SRO thin films have some interesting morphologies depending on their growth conditions. Sample A reveals high quality with a relatively flat surface without any obvious particles This means that no secondary domains were formed, which is consistent with the RSM results. However, the surfaces become rough and harsh when the oxygen pressure decreases. Some tiny protrusions have begun to form on the surface of the film while the atmosphere decreases to $10^{-4}$ Torr, as shown in Figure 3b. There are some uniform muffin-like particles spreading on the flat surfaces of Samples C and D, as shown in Figure 3c,d. The sizes of the muffins strongly depend on the growth pressure. The average diameter and height of Sample C are 0.35 μm and 30 nm, respectively, while those of Sample D are 0.65 μm and 50 nm. This indicates that the formation of the secondary domains as muffin-like regions while the oxygen pressure is below $10^{-4}$ Torr is consistent with the XRD measurements.

SEM-EDS, with the same electron energy, was used to analyze the composition and stoichiometry of the SRO samples. They both contain Sr, O, and Ru (the Ti signal is coming from the substrate). The calculated ratios of Sr/Ru for different samples are quantized and listed. The results of the analysis show that all films exhibited Ru deficiency throughout the working pressure range. As a result, the composition of SRO thin films cannot be directly controlled by the partial pressure parameter. This is consistent with previous findings that growth in an oxygen atmosphere is due to Ru deficiency [6,7,14].

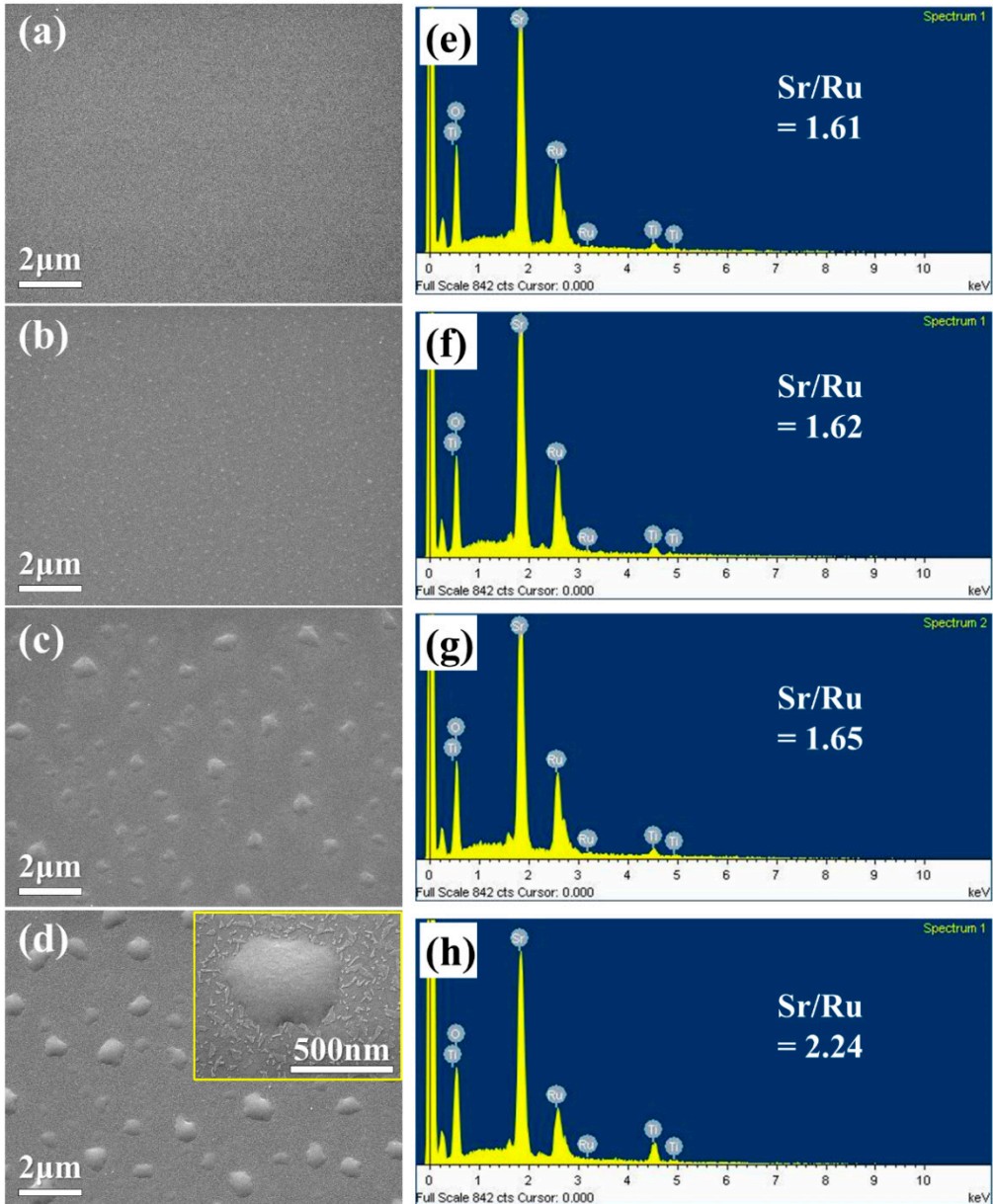

**Figure 3.** (**a–d**) are SEM images of Sample A, Sample C, and Sample D, respectively; (**e–h**) are energy dispersive spectrometer (EDS) images of Sample A, Sample C, and Sample D, respectively. A magnetized image of the muffin-like region is shown in the inset of (**d**). The corresponding atomic ratios of Sr/Ru are also shown on the right side.

Sample D has the highest ratio of 2.24, indicating that more Ru elements run out by reducing the growth ambience. The change in Sr/Ru could be due to the lattice parameter expansion and the conductivity decrease. Keeble et al. showed that *A*-site or *B*-site vacancy in STO perovskite oxide thin films would lead to *c*-axis lattice parameter expansion, which is consistent with our results [31]. The poor electricity can be ascribed to Ru deficiency, which reduces the conduction band hybridization due to there being less overlap between Ru and O orbitals. X-ray photoemission spectroscopy (XPS) was also carried out to analyze the SRO thin films, as shown in Figure S2. The peaks at the binding energies 464.3, 531.7, and 529.2 eV correspond to the Ru $3p_{3/2}$ defect-like oxygen vacancy and SRO (O 1*s*) orbitals, respectively. Technically, it is difficult to determine the exact stoichiometry of oxide thin

films, but the trend of an increase of oxygen vacancies with decreasing oxygen partial pressure can still be obtained, with the result being similar to recent reports [30].

## 3.2. TEM

Sample D was studied using TEM because it showed differences in both structural and electrical properties. As shown in Figure 4a, the cross-section image reveals a 300 nm thickness. The black layer at the top of the thin film is Pt (which is used for SEM measurement), and the bottom corresponds to the STO substrate. The SRO thin film clearly segregates into two different regions, which are, respectively, the muffin-like region with a white color (Region 1) and the thin film region with a gray color (Region 2). The structures of Regions 1 and 2 are analyzed in Figure 4b,c, respectively.

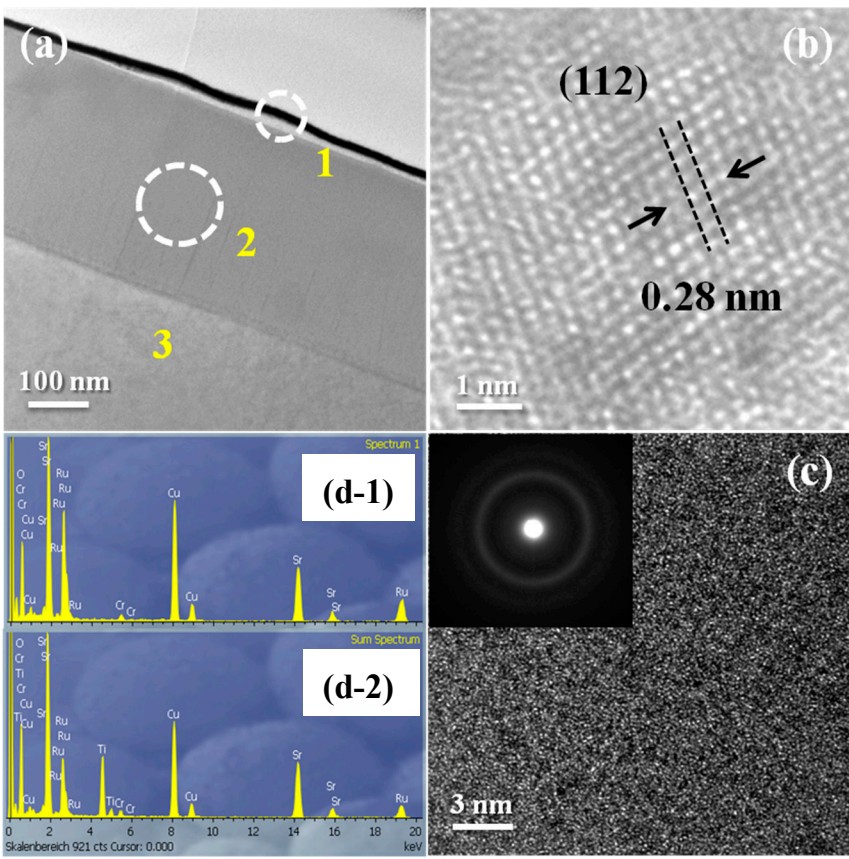

**Figure 4.** (**a**) Cross-sectional high-resolution transmission electron microscope (HRTEM) image of Sample D where Regions 1, 2, and 3 are the muffin region, the thin film region, and the substrate, respectively. (**b**) The magnified TEM of Region 1. (**c**) The magnified TEM of Region 2 and its corresponding selected area diffraction pattern (SAED). (**d**) The selected area TEM-EDS of Regions 1 (**d–1**) and 2 (**d–2**).

The spacing of the (112) planes in the muffin-like region, shown in Figure 4b, is 2.82 Å. It shows little expansion due to the lack of oxygen during growth. However, the diffraction pattern of Region 2 reveals a polycrystalline structure with random orientation. The lattice constant in Region 2 is 2.94 Å. This indicates that Region 1 is closer to the stoichiometry of $SrRuO_3$ compared with Region 2. Figure 4d shows the TEM-EDS of Regions 1 (the upper image) and 2 (the lower image). The signals of Cu and Cr come from the hole-grids, while that of Ti comes from the substrate. Comparing the upper and lower images, the two regions contain apparently different ratios of ingredients, especially the Ru elements. Region 2 contains less Ru than Region 1. The calculated ratio of Ru/Sr is close to one for the muffin-like regions, whereas it is 0.516 for the thin films.

By combining the ingredients with the diffraction patterns in different areas, the muffin-like regions are regarded as SRO. The thin films, however, should be considered to be $Sr_4Ru_2O_9$ [21] for the following reasons: First, the diffraction patterns of thin film around 2.94 Å is close to the (300) peak of $Sr_4Ru_2O_9$, which is the highest diffraction peak in $Sr_4Ru_2O_9$. Second, the ratio of Ru/Sr is near 0.5. Moreover, the $Sr_4Ru_2O_9$ is less conductive than SRO at room temperature [32]. Shin et al. reported that under a strong reduction condition at 700 °C under a high vacuum of $10^{-7}$ Torr, some intermediated SrO-rich phases could form and coexist in the SRO thin film [28]. As a result, the formation of $Sr_4Ru_2O_9$ is not beyond expectation.

The degrees of Ru-deficient SRO thin films are well controlled by using a range of oxygen pressures during deposition. The structure transforms from a perfect epitaxial SRO thin film to a random crystallized SrO-rich phase below $10^{-6}$ Torr. In such a severe condition, SRO splits into the muffin-like regions and the flat regions. The flat regions contain some $Sr_4Ru_2O_9$ components while the muffin-like regions are $SrRuO_3$. SrO-rich phase dominates the structure when both the temperature is high (700 °C) and the oxygen ambience is insufficient ($10^{-7}$ Torr). The formation of SrO-rich phase should influence not only the crystallizations but also the electrical properties of SRO. To analyze the area-dependent-electric properties, we used the scanning probe microscopy technique, which can detect conductive properties in nano-scale size.

*3.3. AFM and C-AFM*

A conductive AFM (C-AFM) was used as a local probe to analyze the electrical properties of the muffin-like regions and the flat regions in Samples C and D. The schematic diagram of the experimental set-up is shown in Figure 5a. Figure 5b shows the surface morphology of Sample C. The average diameter and height of the muffin-like regions in Sample **C** are 0.35 µm and 30 nm, which are consistent with the SEM results.

The current-mapping images of Samples C and D are shown in Figure 5c,d, respectively. The images apparently reveal brighter colors for flat regions and darker ones for muffin-like regions, indicating a different quantity of current flowing through the two regions. For Sample C, the current in the flat region is about 100 nA, while that in the muffin-like region is less than 50 pA. There is an obvious discontinuity between the muffin-like regions and the nearby flat areas since the currents are prevented from transmitting through their interfaces. This might be due to the electron scattering and the barriers at the interfaces. The interfaces involving a structural transform from SRO to $Sr_4Ru_2O_9$ could contain some defects. Sample D shows the same contrast colors between the muffin-like regions and the flat regions, but the current in the flat regions reduces to 5 nA. This might be a result of the lower stoichiometry in Sample D compared with Sample C.

In Figure 5e, the local *I-V* curves of Points 1 and 2 represent the muffin-like region and the flat region for Sample C, respectively, and Points 4 and 6 are those for Sample D. The current of Point 2 increases linearly with the applied voltage. A current of up to 100 nA is observed when applying a voltage less than 0.1 V. Point 6 shows a current of 70 nA as the applied voltage increases to 1 V. It again indicates that there is a decrease in conductivity due to the phase transform from SRO to $Sr_4Ru_2O_9$. As shown in the inset of Figure 5e, Points 1 and 4 show currents lower than 50 pA, even when the voltages are over 1 V. The muffin-like regions embedded in $Sr_4Ru_2O_9$ thin films behave as complete isolators without any leakage currents. In contrast, Sample A has a micro-crystalline area on the surface but is relatively flat compared with Samples C and D, and the current-mapping image on its surface performs almost uniformly except for a few areas, as shown in Figure S3. Not following the step-flow model, the samples in this work are relatively rough, indicating a three-dimensional island growth mode characteristic. The coalescence of grain boundaries and step edges begins to dominate at higher pressures, with the results being similar to the previous reports [7,8], but a better conductive SRO thin film with a stochiometric structure can be conducted in a relatively higher oxygen pressure around $10^{-1}$ Torr. By comparing the C-AFM with the electrical properties listed in Table 1, we can conclude that both the structure and electricity of SRO are strongly affected by the growth ambience. When

the oxygen pressure varies from $10^{-1}$ to $10^{-5}$ Torr, the conductivity decreases slowly due to a lack of Ru elements. In addition, the out-of-plane lattice parameter of SRO expands as a function of reduced oxygen. In this ambience range, the structure seems more sensitive than the electricity. In the second range, when the oxygen pressure is below $10^{-6}$ Torr, the resistivity of SRO increases substantially. The process accompanies a phase transition from SRO to $Sr_4Ru_2O_9$. The secondary phase has an intrinsic resistivity higher than that of intrinsic SRO, which leads to a decrease in the conductivity of SRO thin films.

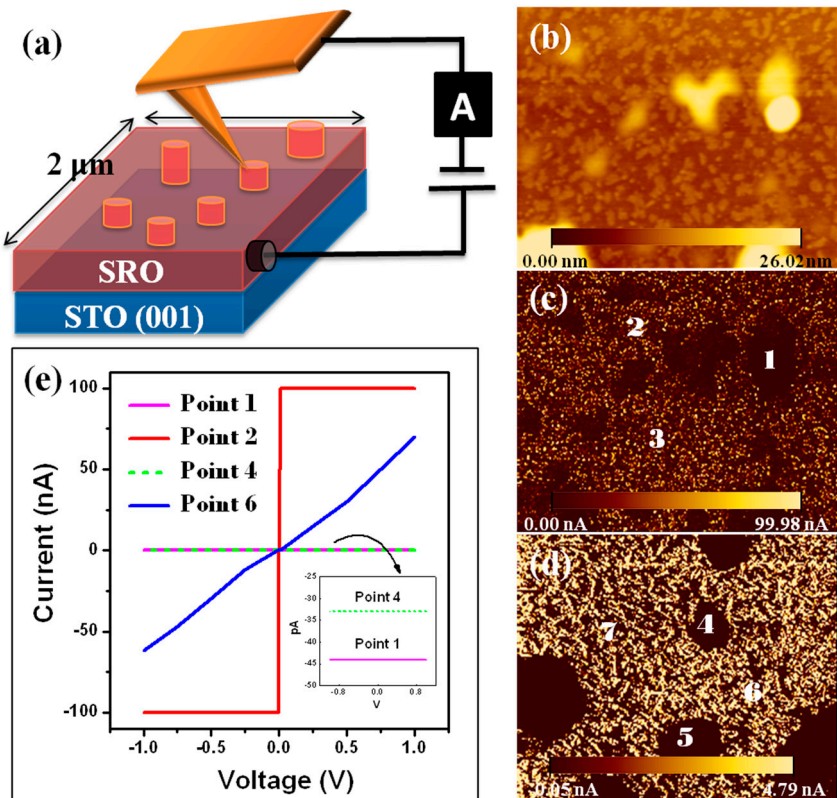

**Figure 5.** (**a**) An experimental set-up of a conductive AFM with a scan scale of 2 μm × 2 μm. (**b**) The surface roughness of Sample C reveals some muffin-like domains. (**c**) and (**d**) show the current mapping images for Samples C and D, respectively. (**e**) The local I-V curves of Points 1, 2, 4, and 6, where the inset figure at the lower and right-hand side is the magnification of the currents of Points 1 and 4, which are on the order of 10–12 Ampere.

## 4. Conclusions

The growth ambience effects on the structure and conductivity of SRO functional oxide films have been investigated in detail by using a series of deposition oxygen pressures. At a pressure of $10^{-1}$ Torr, the epitaxial SRO thin film reveals a low resistivity of about 200 μΩ-cm and an excellent crystallization with a sharp FWHM of 0.071° and 0.047° for the x-ray rocking curve and the ɸ-scan, respectively. The lattice parameter increases gradually below $10^{-4}$ Torr, and resistivity rises significantly above $10^{-4}$ Torr as well. The causes are ascribed to the loss in Ru elements and a phase transformation from SRO to $Sr_4Ru_2O_9$. The corresponding microstructures and their conductive properties have also been investigated by utilizing electron microscopy and conductive force microscopy.

**Supplementary Materials:** The following are available online at http://www.mdpi.com/2079-6412/9/9/589/s1, Figure S1: The growth ambience dependent XRD patterns around surface normal of SRO thin films on STO (001) deposited with the oxygen pressure of $10^{-1}$, $10^{-4}$, $10^{-6}$, and $10^{-7}$ Torr, respectively, Figure S2. (a) and (b) are X-ray photoemission spectra of Ru $3p$ core-level and O $1s$ for SRO thin film samples, respectively, Figure S3. (a) and

(b) The surface topography and the current mapping images for samples A, respectively. (c), (d), (e) and (f) The corresponding local I-V curves of region A, B, C, and D.

**Funding:** This research received no external funding.

**Acknowledgments:** H.-M.C. would like to thank C.Y. Tsai., C. Kao., and S. Yang for operating the PLD system. The input of W.S. Hsu, who provided C-AFM support, is also appreciated.

**Conflicts of Interest:** The authors declare no conflict of interest.

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
