# Peer review of "Influence of the Growth Ambience on the Localized Phase Separation and Electrical Conductivity in SrRuO3 Oxide Films"

_coatings, doi:10.3390/coatings9090589_

Round 1

Reviewer 1 Report

Coatings-586663 

Title: Influence of the Growth Ambience on the Localized Phase Separation and Electrical Conductivity in SrRuO3 Oxide Films

Author: Hsin-Ming Cheng

The authors present the properties of thin films of SrRuO3 grown on STO. The manuscript is well written, the state-of the art is very well addressed; the details of the work and the interpretations are convincing.

There are a few issues to be addressed before publication:

The results, especially AFM-C AFM, should be introduced for sample A too.

Minor issues:

More details about the deposition parameters should be introduced, for example laser spot size or laser fluence; heating/ cooling rate; oxygen pressure during cooling, substrate - target distance. Line 126: please remove "small" from "some uniform small muffin-like particles" as there is no recognized classification for particle size domain as small or big.

There are a few typing errors, but the text is fluent and easy to read.

In my opinion, the topic of the paper makes the work useful and appropriate for Coatings.

Author Response

Response to the comments of referee 1:

1. The authors present the properties of thin films of SrRuO3 grown on STO. The manuscript is well written, the state-of the art is very well addressed; the details of the work and the interpretations are convincing.

All authors deeply appreciated reviewer’s encouragement.

2. There are a few issues to be addressed before publication:

The results, especially AFM-C AFM, should be introduced for sample A too.

The AFM and C-AFM data of sample A have been upgraded in the Supplementary Information and comparison with sample C& D was described in the main text.

3. Minor issues:

More details about the deposition parameters should be introduced, for example laser spot size or laser fluence; heating/ cooling rate; oxygen pressure during cooling, substrate - target distance.

Line 126: please remove "small" from "some uniform small muffin-like particles" as there is no recognized classification for particle size domain as small or big.

There are a few typing errors, but the text is fluent and easy to read.

In my opinion, the topic of the paper makes the work useful and appropriate for Coatings.

The description of “Experiment Part” has been upgraded in detail according to the comments in the main text. The typewriting mistakes were already corrected conservatively. The manuscript has been edited for consistent quality of the content by a native English editor and the flaws in grammar have been revised. We consider the article would suitable for reading. All authors deeply appreciated reviewer’s suggestion.

Reviewer 2 Report

SRO is a very well studied metallic complex oxide. It has been known for a long time already that it is able to grow in the step-flow growth mode in PLD. This growth mode is exceptional and produces atomically smooth surfaces, which is of critical importance for further epitaxial growth of other oxides on top of SRO as well as for the ability to fabricate and study ultrathin layers. See, for example, DOI: 10.1103/PhysRevLett.103.057201, 10.1103/PhysRevB.83.193401, 10.1063/1.3001932, 10.1063/1.1640472, 10.1103/PhysRevLett.95.095501, as well as references therein and citing papers. This behavior is peculiar for SRO, and the reasons for it were discussed in the literature. This very important fact about SRO is not mentioned in the manuscript. Overall, quite old literature is cited. Both are clear shortcomings. The author should more carefully review what is known about SRO and place the reported research in a relevant context. With the knowledge on PLD growth of SRO gathered to date, the aim of the reported research and the value of the presented data are not clear.    

Author Response

Response to the comments of referee 2:

1. SRO is a very well studied metallic complex oxide. It has been known for a long time already that it is able to grow in the step-flow growth mode in PLD. This growth mode is exceptional and produces atomically smooth surfaces, which is of critical importance for further epitaxial growth of other oxides on top of SRO as well as for the ability to fabricate and study ultrathin layers. See, for example, DOI: 10.1103/PhysRevLett.103.057201, DOI:10.1103/PhysRevB.83.193401, 10.1063/1.3001932, 10.1063/1.1640472, 10.1103/PhysRevLett.95.095501,

as well as references therein and citing papers. This behavior is peculiar for SRO, and the reasons for it were discussed in the literature. This very important fact about SRO is not mentioned in the manuscript. Overall, quite old literature is cited. Both are clear shortcomings. The author should more carefully review what is known about SRO and place the reported research in a relevant context.

All authors are grateful to the review who is expert of SRO for such earnest advice. Based on the articles recommended by the review and the relevant professional background, we update the part of introduction to strengthen the SRO knowledge and current research progress in this manuscript. The corresponding reference and related crucial achievements have been cited (Reference 17-22) and described in the main article, respectively.

2. With the knowledge on PLD growth of SRO gathered to date, the aim of the reported research and the value of the presented data are not clear

As far as previous research is concerned, it has indeed laid a good foundation for SRO coating technology, for example the “Step Flow Growth”. However, the control of the atmosphere, especially the oxygen ambience plays an essential role in the aspects of defects, stoichioemtry, oxidations, reductions, and phase transformations. The changes in the microstructure of surfaces and electrical properties of SRO with the growth environment was rarely mentioned. As a result, in this study we specifically study microstructures, with particular emphasis the relationship on oxygen atmosphere and film morphology (especially the phase transform from ruthenium deficiency, nonstoichiometric domain, and corresponding electrical conductivity) for which the interesting behaviors of SRO films I think was not be disclosed similarly as our work before.

Reviewer 3 Report

In this research, the author has synthesized perovskite SrRuO3 (SRO) epitaxial thin films grown on SrTiO3 (001) using pulsed laser deposition with varying ambient oxygen pressure. The author found that the electrical resistivity of the high Oxygen ambient pressure increases by about two orders of magnitude as compared to that of the low Oxygen pressure samples.

In the PLD method, the oxygen partial pressure has a decisive effect on the physical properties of SRO films because the working pressure of oxygen has a direct effect on the characteristics of the plume ablated from metal oxide targets. The effect of the oxygen partial pressure on the structural and electrical properties of epitaxial SrRuO3 thin films grown by PLD has already been studied by many groups recently.

Physica B: Condensed Matter 572 (2019) 190–194; https://doi.org/10.1016/j.physb.2019.07.038 J. Appl. Phys. 120, 235108 (2016); https://doi.org/10.1063/1.4972477 J. Journal of Appl. Phys., 35 (1), 12A; https://doi.org/10.1143/JJAP.35.6212 J. Appl. Phys. 97, 103525, (2005). https://doi.org/10.1063/1.1909284

My question to the author is what the novelty in this work?

My other specific comments on the manuscript are:

There are a lot of grammatical and typos in the paper. I recommend that the author should proofread the article. To eliminate the thickness dependence and to guarantee the intrinsic physical properties of films, the thicknesses of all the SRO samples should be kept constant. The author has not mentioned anything about the thickness of the samples. What was the target to substrate distance? The author has grown four samples (A-D) but discusses the RSM and XRD data of samples A and C only. Similarly, SEM and AFM of all the samples are not considered. I suggest that the author should discuss all the four samples in his results and discussion. Stoichiometry near the SRO surface changes, mostly due to ruthenium loss or a segregation process, resulting in a mixed oxidation state from metallic to fully oxidized and hence affecting the electrical properties. Hence, the author should consider imperfections and high reactivity of its surface region. I suggest that the author should study the electronic properties by, x-ray absorption near-edge structure (XANES) or X-ray photoelectron spectroscopy (XPS) measurements. The ToF SIMS can also be used to check whether the stoichiometry of the film is constant or not.

 Author Response

1. In this research, the author has synthesized perovskite SrRuO3 (SRO) epitaxial thin films grown on SrTiO3 (001) using pulsed laser deposition with varying ambient oxygen pressure. The author found that the electrical resistivity of the high Oxygen ambient pressure increases by about two orders of magnitude as compared to that of the low Oxygen pressure samples.

In the PLD method, the oxygen partial pressure has a decisive effect on the physical properties of SRO films because the working pressure of oxygen has a direct effect on the characteristics of the plume ablated from metal oxide targets. The effect of the oxygen partial pressure on the structural and electrical properties of epitaxial SrRuO3 thin films grown by PLD has already been studied by many groups recently.

Physica B: Condensed Matter 572 (2019) 190–194; https://doi.org/10.1016/j.physb.2019.07.038

Appl. Phys. 120, 235108 (2016); https://doi.org/10.1063/1.4972477 Journal of Appl. Phys., 35 (1), 12A; https://doi.org/10.1143/JJAP.35.6212 Appl. Phys. 97, 103525, (2005). https://doi.org/10.1063/1.1909284

My question to the author is what the novelty in this work?

We thank the review’s earnest advice and careful reminder. Based on these relative articles that recommended by the reviewer, we have already read and cite in this article.

In the main text we also supplement the following statement:

“However, the previous reports revealed the resistivity of SRO thin films as entire thin film, namely the macro-scale measurement, there are few investigations that mentioned about the microstructural and local electrical properties of SRO with various growth ambience. As a result, to understand the complexity between the process and the conductivity that can be conducted practical application in the future. The subject related to the growth ambience of SRO thin films needs to be diligently discussed.

In this study, we control the oxygen pressure from 10-1 to 10-7 Torr for systematically studying the influence of growth ambience on the structure and conductivity of SRO thin films. The crystallization and the corresponding conductivity of SRO thin films vary significantly as a function of the ambience environments. Furthermore, the variations of crystallization from the deficiency of Ru elements, surface morphology, degree of stoichiometry, and the microstructure caused from phase transformation in SRO thin films are also investigated.”

 Comprehensively speaking, in this study we specifically study microstructures, with particular emphasis the relationship on oxygen atmosphere and film morphology (especially the phase transform from ruthenium deficiency, microstructure, and corresponding electrical conductivity) for which the interesting behaviors of SRO films I think was

2. My other specific comments on the manuscript are:

There are a lot of grammatical and typos in the paper. I recommend that the author should proofread the article. To eliminate the thickness dependence and to guarantee the intrinsic physical properties of films, the thicknesses of all the SRO samples should be kept constant. The author has not mentioned anything about the thickness of the samples. What was the target to substrate distance?

The thickness of each SRO thin films was controlled to around 300nm and target to substrate distance of ~55 mm.

The manuscript has been edited for consistent quality of the content and the flaws in grammar have been revised. We consider the article would suitable for reading.

3. The author has grown four samples (A-D) but discusses the RSM and XRD data of samples A and C only. Similarly, SEM and AFM of all the samples are not considered. I suggest that the author should discuss all the four samples in his results and discussion. Stoichiometry near the SRO surface changes, mostly due to ruthenium loss or a segregation process, resulting in a mixed oxidation state from metallic to fully oxidized and hence affecting the electrical properties. Hence, the author should consider imperfections and high reactivity of its surface region. I suggest that the author should study the electronic properties by, x-ray absorption near-edge structure (XANES) or X-ray photoelectron spectroscopy (XPS) measurements. The ToF SIMS can also be used to check whether the

  All authors thank reviewer’s useful advices. The SEM, and AFM parts are already updated. The more complete information is supplied to the main text and supporting information as well. X-ray photoemission spectroscopy (XPS) were also carried out to analyze the trend as an increase of oxygen vacancies with decreasing oxygen partial pressure. The additive spectra are supported in supporting information.

    For the extra check, we also apply the beam time of Synchrotron and seek assistance with x-ray absorption near-edge structure (XANES) and ToF SIMS, respectively, which need more time to process the reservation. We regret that cannot obtain the potential data from these analysis timely because of the time limitation. The complete analytical methods will be used more widely in our forthcoming studies.

Round 2

Reviewer 2 Report

In the revised version, introduction to the topic and to the previous work performed on the SrRuO thin films improved, and now it is acceptable. The only factual correction that should be made. The authors state that "in the past ten years, SRO thin films have been proposed that could be successfully deposited on vicinal substrates such as STO through heteroepitaxial step-flow growth [23-26]". However, ref [23] is dated to 2004 and [13] to 1996. Hence, this growth has been studied for almost 25 years by now.

Otherwise, the manuscript can be published in the present form in Coatings.     

Author Response

Comments and Suggestions for Authors

In the revised version, introduction to the topic and to the previous work performed on the SrRuO thin films improved, and now it is acceptable. The only factual correction that should be made. The authors state that "in the past ten years, SRO thin films have been proposed that could be successfully deposited on vicinal substrates such as STO through heteroepitaxial step-flow growth [23-26]". However, ref [23] is dated to 2004 and [13] to 1996. Hence, this growth has been studied for almost 25 years by now.

Otherwise, the manuscript can be published in the present form in Coatings.

We thank the review’s careful reminder. The modification has bee done in the introduction section.

Reviewer 3 Report

Since the authors have considered all of my concerns in their revised manuscript, it can be accepted to be published in the journal Coatings

Author Response

Comments and Suggestions for Authors

Since the authors have considered all of my concerns in their revised manuscript, it can be accepted to be published in the journal Coatings

The author and team members deeply appreciate the reviewer for the efforts to encourage our research and suggestions for improving the quality of this manuscript.